# Identifying plausible historical scenarios for coupled lake level and seismicity rate changes: The case for the Dead Sea during the last two millennia.

**Mariana Belferman[1], Amotz Agnon[2], Regina Katsman[1] and Zvi Ben-Avraham[1]**

*[1] The Dr. Moses Strauss Department of Marine Geosciences, Leon H. Charney School of Marine Sciences, University of Haifa, Mt. Carmel, Haifa 3498838, Israel.*

*[2] The Fredy & Nadine Herrmann Institute of Earth Sciences, The Hebrew University of Jerusalem, Jerusalem 9190401, Israel*

Mariana Belferman: mkukuliev@gmail.com (corresponding author)

Amotz Agnon: amotz@huji.ac.il

Regina Katsman: rkatsman@univ.haifa.ac.il

Zvi Ben-Avraham: zviba@post.tau.ac.il

**ABSTRACT**

Studies of seismicity induced by water level changes in reservoirs and lakes focus typically on well-documented contemporary records. Can such interactions be explored on a historical time scale when the two data types suffer from severe uncertainties stemming from the different nature of the data, methods, and resolution? In this study, we show a way to considerably improve the correlation between interpolated records of historical Dead Sea level reconstructions and discrete seismicity patterns in the area, over the period of the past two millennia. Inspired by the results of our previous study, we carefully revise the historical earthquake catalog in the Dead Sea, to exclude remote earthquakes, and include small local events. For addressing the uncertainties in

lake levels, we generate an ensemble of random interpolations of water level curves and rank them
by correlation with the historical records of seismic stress release. We compute a synthetic catalog
of earthquakes applying a Mohr-Coulomb failure criterion. The critical state of stress at
hypocentral depths is achieved by static poro-elastic deformations incorporating the effective
normal stress change (due to the best-fit water level curve) superimposed on the regional strike-
slip tectonic deformations. The earthquakes of this synthetic catalog show an impressive
agreement with historical earthquakes documented to damage Jerusalem. We refine the seismic
catalog by searching for small local events that toppled houses in Jerusalem; including all local
events improves the correlation with lake levels. We demonstrate for the first time a high
correlation between water level changes and the recorded recurrence intervals of historical
earthquakes.

## KEYWORDS

Induced seismicity; Seismic recurrence interval; Water level changes; Effective stress; Dead Sea;
poro-elastic response.

## INTRODUCTION

Earthquakes induced by water level changes in lakes and reservoirs has been a focus of

seismic investigations around the world (e.g. Simpson et al., 1988; Pandey and Chadha, 2003;
Durá-Gómez and Talwani, 2010). Triggering is attributed to a drop in the effective normal stress
at a fault, induced by water level change at the overlying lake's bed (Simpson et al., 1988; Durá-
Gómez and Talwani, 2010; Hua et al., 2013b; Gupta, 2018). This kind of triggering may be
particularly significant for areas with moderate and low tectonic strain accumulations (Pandey and
Chadha, 2003; Gupta, 2018), such as the Dead Sea fault in the Middle East (e.g., Masson et al.,

2015).

Seismic activity due to water level change was observed beneath artificial reservoirs

immediately after their first filling (e.g., Simpson et al., 1988; Hua et al., 2013 a). It also appeared
after several seasonal filling cycles (Simpson et al., 1988; Talwani, 1997), explained by diffusion
of pore pressure to the earthquake's hypocentral depth via the fault (Durá-Gómez and Talwani,
2010). In addition, reservoir-induced seismicity sometimes manifests itself at long distances away
from the reservoir (e.g., at 35 km, Durá-Gómez and Talwani, 2010). The correspondence of this
kind of contemporary seismicity to water level change is usually identified based upon real-time
data.

Alternatively, on a much longer time scale, changing seismic activity may also be associated

with water level changes in historical water bodies (e.g., the Dead Sea, since 2 ka, Fig. 1A, in the
Appendix, which occupies the tectonic depression along the Dead Sea fault). Water level hikes of
~15 m, characteristic for time intervals of centuries to millennia, were analyzed in Belferman et
al. (2018) and shown to be able to moderate the seismicity pattern at the Dead Sea fault.

However, reconstruction of fluctuations in historical lake levels and the concurrent

seismicity are both subject to significant uncertainties. They stem from the differing nature of the
data gathered on these two phenomena and thus deserve special consideration. Earthquake dating
can be quite precise, and its accuracy is verified when different historical sources show consensus
(Guidoboni et al., 1994; Guidoboni and Comastri, 2005; Ambraseys, 2009). Assessment of the
extent of damage (hence earthquake magnitude), similarly requires such a consensus between the
different data sources. Sediment records can help to calibrate the analysis of the historical evidence
(Agnon, 2014; Kagan et al., 2011). Such records can be tested by trenching (Wechsler et al., 2014;
Marco and Klinger, 204; Lefevre, 2018). However, in many cases earthquake epicenter can be
imprecise or not even known. Consequently, considerable uncertainty pertains to the historical
catalog of earthquakes related directly to the Dead Sea.
By contrast, historical water level records are quite precise elevation wise, as they are
obtained from different points around the lake (Bookman et al., 2004; Migowski et al., 2006).
However, water level dating could have an error of about ±45 yr, as estimated from the radiocarbon
dating of shoreline deposits in fan delta outcrops (Bookman et al., 2004). This may underestimate
the actual dating uncertainty due to reworking of organic matter, sometimes re-deposited a century
or more after equilibration with the atmosphere (Migowski et al., 2004). In addition, the entire past
bi-millennial Dead Sea level record is constrained by less than twenty "anchor points" (the data
obtained by the dating collected from surveyed paleo-shorelines, Bookman et al., 2004). Therefore,
its continuous reconstruction, as suggested in the literature (Migowski et al., 2006; Stern, 2010),
usually takes different forms within the acceptable limits dictated by the evidence,
geomorphological (Bookman et al., 2004) and limnological (Migowski et al., 2006). A challenging
uncertainty for our study arises from the interpolations required for periods when the available
data do not constrain the water levels.
In this article, we take advantage of the correlation between the historical water level (WL)
reconstructions at the Dead Sea and seismicity patterns in the area over the past two millennia. We
demonstrate for the first time that plausible scenarios for the lake level history can fit very well
the record of the historical earthquake recurrence intervals (RIs). Based on the correlation between
these phenomena, we offer an alternative explanation regarding the triggering of earthquakes in
the area of the Dead Sea.

## METHODS

To investigate the relationship between an accurate but discrete chronology of earthquakes and the continuous water WL change, we first explore the space of possible WL histories by a statistical approach. We generate an ensemble of WL curves (based on the anchor points, Bookman et al., 2004), while remaining within the limits dictated by climatic and morphological constraints (Bookman et al., 2004; Migowski et al., 2006; Stern, 2010), by using a random number generator.

In our analysis we associate all the historical earthquakes presented (Table 1A,2A in Appendix) with rupture of the strike-slip faults, which agree with our modeling approach. Hence, the major strike-slip faults constituting the plate boundary (Lower Jordan fault, Dead Sea Lake fault and Northern Arava) could be affected by Dead Sea WL changes. Therefore, our study covers the area within this distance

**A best fit random method of WL curve prediction**

The compilation of WL curves of the Dead Sea for the last two millennia from three recent publications (Bookman et al., 2004; Migowski et al., 2006 and Stern 2010) is presented in Figure 1A by dashed curves. Generally, the differences between all dashed curves at anchor points is included within an error limit of ±45 yr as indicated by error bars, with an exception of the anchor point dated to 1400 CE (Bookman et al., 2004) for which Migowski et al. (2006) and Stern (2010) suggested a higher WL. Nevertheless, each hypothetical WL curve is forced to pass through all anchor points provided by Bookman et al. (2004) except for one, at around 500 CE. The WL drop around this time, according to Migowski et al. (2006) and Stern (2010), occurred later than was originally suggested by Bookman et al. (2004) (Figure 1A). Because this shift is within the permissible error limits (±45 yr), this anchor point is shifted to the left (+40 yr). In addition, the

WL determined on the curve edges of the studied bi-millennial time interval was defined by
additional two anchor points, through which the estimated WL curve passed according to all three
references. In total, we have 13 anchor points. Between each pair of points, the trends in the WLs
are constrained by the sedimentary facies (Migowski et al., 2006) that specify the edge points of
the interval as the extrema for the acceptable WL variation.
However, within the largest interval between the anchor points (600 - 1100 CE), the field
studies (Migowski et al., 2006; Stern, 2010; Bookman et al., 2004) constrained the WL to be lower
than the extrema at the edges of that interval. For this period, the WL was randomly interpolated
between the higher (e.g., Migowski et al., 2006) and lower (e.g., Stern, 2010) bounds. To maintain
a monotony of the WL variation (required by the facies analysis of Migowski et al.), a moving
average filtered the random noise between every pair of the anchor points. Accounting for the
above-mentioned limits, and setting a ten-year step, the model has generated 10 million WL curves
for the last bimillennial interval, using a uniformly distributed random number generator.
We test for linear correlation between the RIs of the widely recorded moderate-to-large
(M>5.5) historical earthquakes available from the literature (Table 1 and the text description in
Appendix), and the WL interpolations (as in Figure 9 in Belferman et al., 2018); and evaluate the
values of the Pearson product-moment correlation coefficient, R (Figure 1B). We use this statistic
for evaluating the suitability of each randomly interpolated WL curve for our analysis, for
identification and elimination of any outliers, and for studying the behavior of the entire ensemble
of the curves generated.
**The earthquake simulation algorithm**

The most suitable WL curve suggested by this correlation (discussed in the results section

below), was used to generate a "synthetic" earthquake catalog based on the algorithm described in
this section. Effective (normal) poroelastic stress change due to the WL change is superimposed
on the tectonic stress accumulated consistently with the slip rate since the preceding seismic event,
and synthetic earthquakes are simulated using a Coulomb failure envelope and a Mohr circle (e.g.
Jaeger et al., 2009). A vertical strike-slip fault below the lake/reservoir bed is assumed (simulating
a Dead Sea fault), embedded in the 2D (plain strain) geometry of the upper crust (Belferman et al.,
2018). Tectonic horizontal strike-slip displacements across the fault are approximated by a simple
shear approach with no normal strain component.

In the poroelastic part of the model, horizontal stress change normal to the strike slip fault

produced by the WL change, is calculated under a uniaxial (vertical) strain condition (Eq.10b in
Belferman et al., 2018). This is applicable to a post-diffusion stage: i.e., when pore pressure at
hypocentral depth equilibrates with the lake's bed. An array of the effective horizontal normal
stress changes, $\Delta\sigma_i'$, at the fault, induced by the water load change at the lake's bed, $p_{s_i}$,
corresponds to the array of the WL change, $\Delta h_i (i = 1, 2, \dots 2000)$ over the interpolated WL curve,
Figure 1D:

1.            $\Delta\sigma_i' = \frac{1-2\nu}{1-\nu}(\beta - 1)p_{s_i}$

(Eq. 10b in Belferman et al., 2018). Here $\beta$ is Biot's coefficient and $\nu$ is the Poisson's ratio, $p_{s_i} =$
$\rho g \Delta h_i$, where $\rho$ is the density of water and $g$ is the acceleration of gravity.

A radius and a center location of the Mohr circle change as a function of the tectonic

deformations and WL changes, correspondingly, eventually reaching a failure envelope that
simulates an earthquake. The model uses a Byerlee's law envelope (Byerlee, 1978) to define a
residual strength of a seismogenic zone at the fault immediately after the earthquake (Belferman
et al., 2018 for more detail). Since the effective stress upon the onset of an earthquake is specified
by a high failure envelope and the effective stress following the slip is given by Byerlee's law
(e.g., Belferman et al., 2018), the model is time-predictable. The stress drop, at least in the
nucleation zone of a single-fault model, is expected to be proportional to the RI.

A starting point of the simulations is the date of the first historical earthquake (33CE,

Table 1 in the Appendix) from the bi-millennial time interval studied. The simulation
incrementally proceeds with time over the WL curve generated (as above) under the accumulating
tectonic stress. After each stress release, the time to the next earthquake, $\Delta t$, is calculated from the
solution of the Mohr-Coulomb failure criterion for a strike-slip tectonic regime and a WL change,
$\Delta h_i$, characteristic of the Dead Sea fault (Belferman, et al., 2018):

2.        $$(\tau_i - \tau_0)^2 + \left(\sigma_i - (\sigma_0 + \Delta\sigma_i')\right)^2 = \left(R_0 + \Delta\tau_{xy_i}\right)^2$$

$$\tau_i = C + tan(\varphi)\sigma_i$$
assuming that $\Delta\tau_{xy_i} = \frac{Ccos(\varphi)}{t_{RI}}\Delta t$ is the tectonic shear stress accumulated consistently with slip-
rate at the strike-slip fault during the period $\Delta t$ (time passed since the last earthquake), $C$ is
cohesion, $\varphi$ is an angle of internal friction, $\sigma_0$ and $\tau_0$ are the coordinates of the Mohr circle center
immediately after the earthquake and $R_0$ its radius, $t_{RI}$ is the reference RI corresponding to the
minimal WL.

For each time step, the algorithm determines whether there is a single solution, or two, or

nil. A case of no solutions means that the Mohr circle is yet to reach the failure envelope, as the
accumulating tectonic stress and the WL increase are still insufficient. The system of Eq. 2 may
have a single solution when the failure criterion is met at the end of some timestep, or two solutions
when it is met before the end of the timestep. A case of two solutions is rounded down to a case
of a single solution if a time step (one year) is small compared to the earthquake RI (several
hundreds of years).

This solution of Eq.2 yields a RI as a function of the effective normal horizontal stress

change, $\Delta\sigma_i'$ (Belferman et al., 2018):

3.        $$RI = \Delta t = (C + tan(\varphi)\Delta\sigma_i')\frac{t_{RI}}{C}$$

where $t_{RI}$ is the reference $RI$ corresponding to the minimal WL, $C$ is cohesion, $\varphi$ is an angle of
internal friction. From this formula, an array of earthquake dates is obtained.
Substitution of Eq.1 into Eq.3, yields a linear dependence of a simulated $RI$ on a WL change, $\Delta h_i$,
evolving with time.

4.        $$RI = t_{RI} + \frac{tan(\varphi)}{C}\frac{1-2\nu}{1-\nu}(\beta - 1)\rho g t_{RI}\Delta h_i$$

A tectonic slip-rate is set at 5 mm/yr (e.g. Hamiel et al., 2018; Hamiel and Piatibratova, 2019;
Masson et al., 2015). Coefficients for the simulations were previously determined in Belferman et
al. (2018). Note that the cohesion, $C$, is not a-priory known, hence it is fixed by the empirical
correlation between WL and RI for a given lake level history considered. Its value, $C = 0.08Mpa$,
and a reference RI, $t_{RI} = 300yr$, were adjusted numerically for a WL curve, providing the average
RI of 144 yr over the modeled period of two millennia justified by historical, archaeological, and
geological data (Agnon, 2014).

## RESULTS

Ten most suitable WL curves are identified out of the 10M set of WL randomly generated curves ("ensemble") by the Pearson product-moment correlation test. The values of the correlation coefficients, R, for the entire ensemble are distributed normally around R=0.63 (Figure 1B) with a standard deviation of $\sigma = 0.076$. The ten most suitable WL curves ordered by their correlation coefficients, R, are presented in Figure 2.

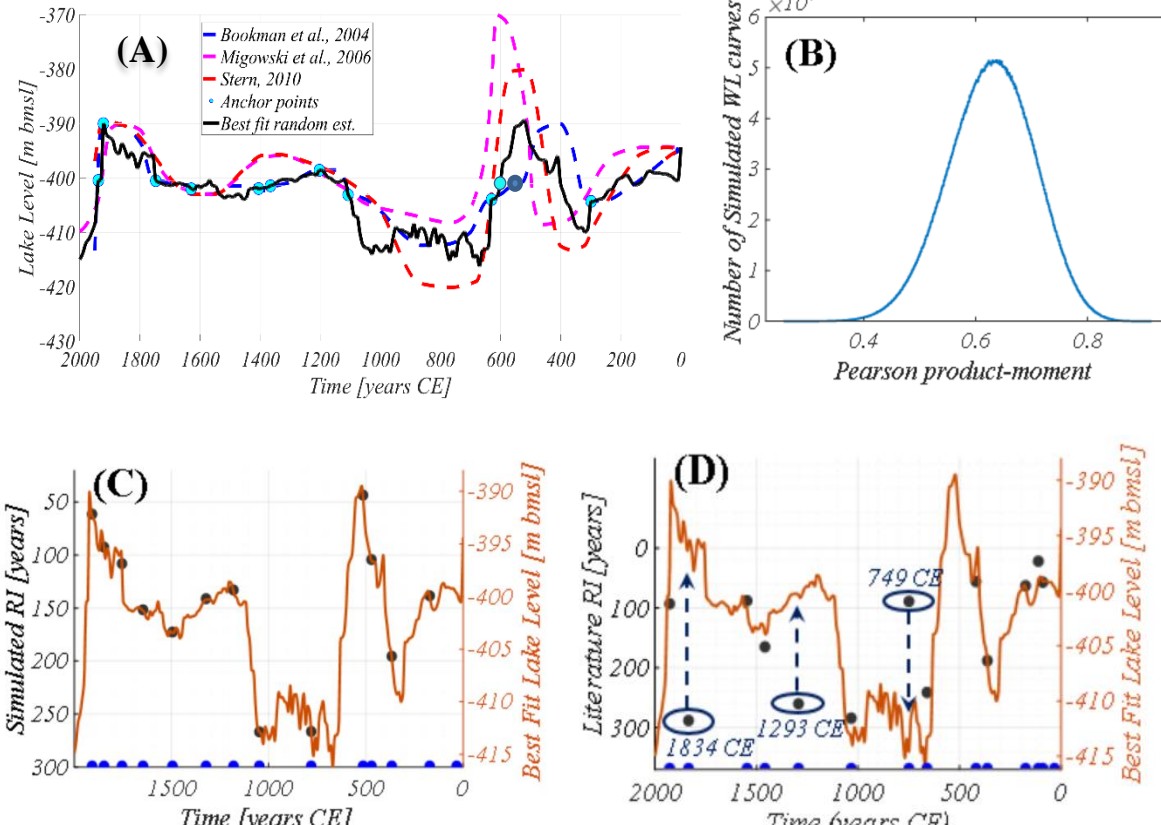

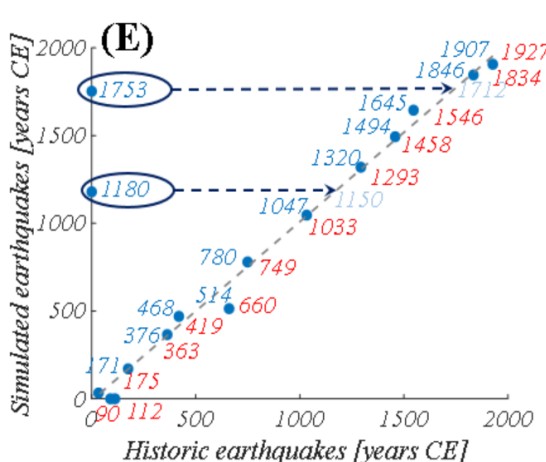

**Figure 1:** (*A*) *The Dead Sea WL reconstructions for the last two millennia. The dashed curves are suggested by the literature sources. Turquoise anchor points follow Bookman et al. (2004) used in WL interpretation, while one point (in dark blue) is shifted to left in error interval of ±45 yr. Solid, black line water curve is suggested by this study. (B) Distribution of Pearson's product-moment correlation coefficient of randomly interpolated WLs and RIs of historic earthquakes. Normal distribution results from 10M random WLs reconstructions. (C) and (D): Orange curve represents the best fit random WL curve vs. simulated and historic RIs, correspondingly. The blue dots mark the dates of the seismic events, while the black dots indicate the recurrence interval between these events. For optimal visualization of the correlation, the degree of scaling freedom for the RI axis was set for these figures. (E) Dates of historic vs. simulated earthquakes based on the suggested best fit WLs curve (Figs.C,D), are compared.*


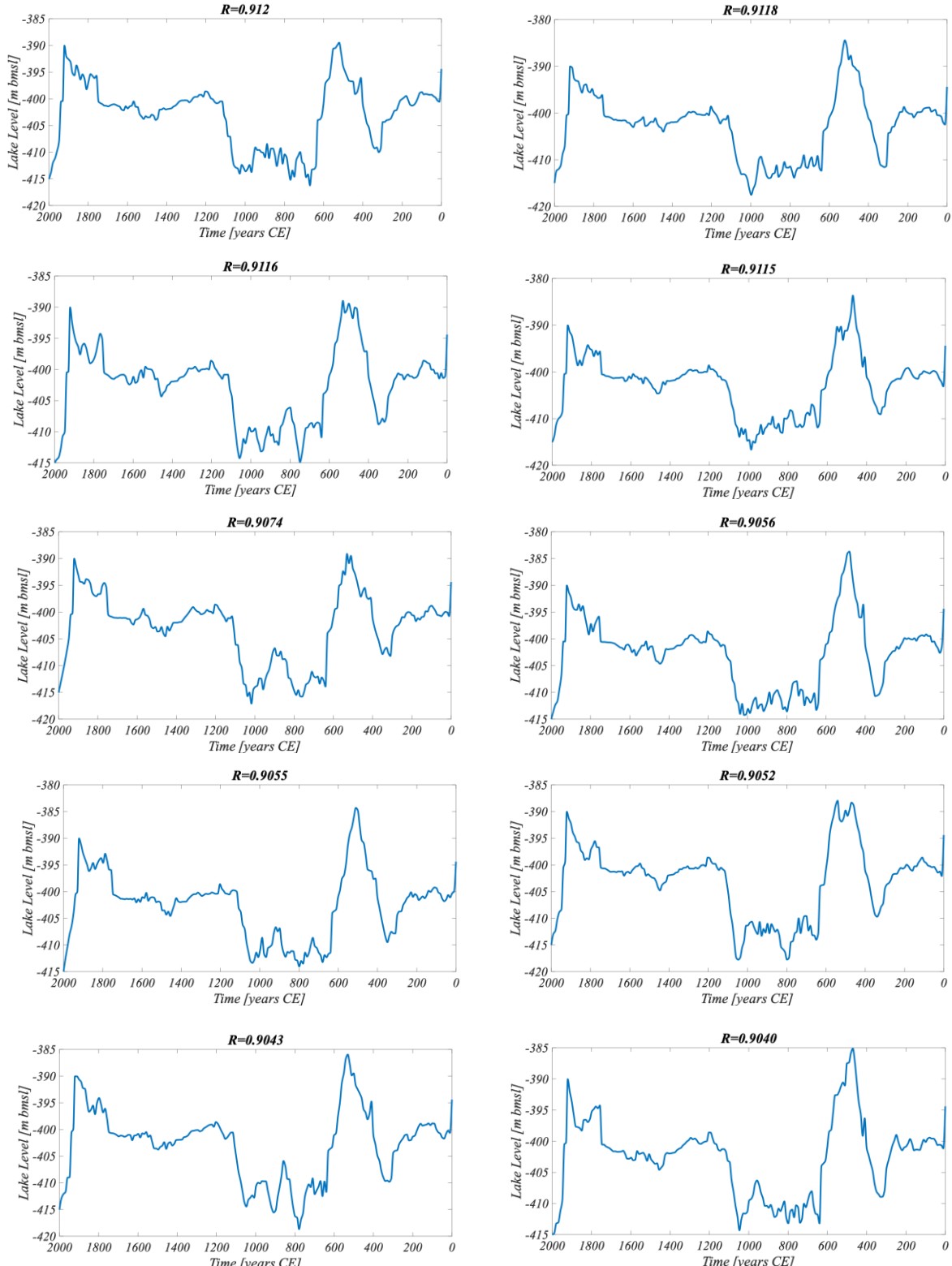


**Figure 2:** *Ten most suitable WLs identified out of the 10M randomly generated by the Pearson product-moment correlation test.*


Three outliers from the thirteen RIs of the widely recorded historic earthquakes (749 CE,
1293 CE and 1834 CE in Figure 1) were identified and reevaluated (explanation in Appendix). A
curve with the highest Pearson coefficient of R=0.912 was chosen from the correlation between
the RIs of the revised historic catalog and the randomly generated WLs (Figure 2). This correlation
can be specified by a linear prediction function:

5.          $RI = -5442 - 14WL$

where RI is given in years and WL in meters. In addition, a synthetic earthquake history including
14 seismic events was simulated from the best fit randomly interpolated WL curve with R=1
specified above. The synthetic RIs can be approximated based on the WLs using the linear
relationship Eq.4 ):

6.          $RI = -3840 - 10WL$

The dates of the simulated synthetic earthquakes are presented, versus the dates of the historical
earthquakes from the literature (Table A1, Appendix) in Figure 1E.
The synthetic earthquake stress history is presented in Figure 3. The effective horizontal
normal stress change, $\Delta\sigma_i'$, (Figure 3A) linearly depends on the WL (Eq.1.), and as expected,
follows its variability. The tectonic shear stress change, $\Delta\tau_{xy}$, drops to zero after the accumulated
shear stress is released by the strike-slip earthquake (Figure 3B). Less shear stress is required to
induce the earthquake when the change in WL is larger (Figures 3A,3B), modeled with Mohr-
Coulomb failure criteria (Figure 3C) (explained also in Belferman et. al., 2018).

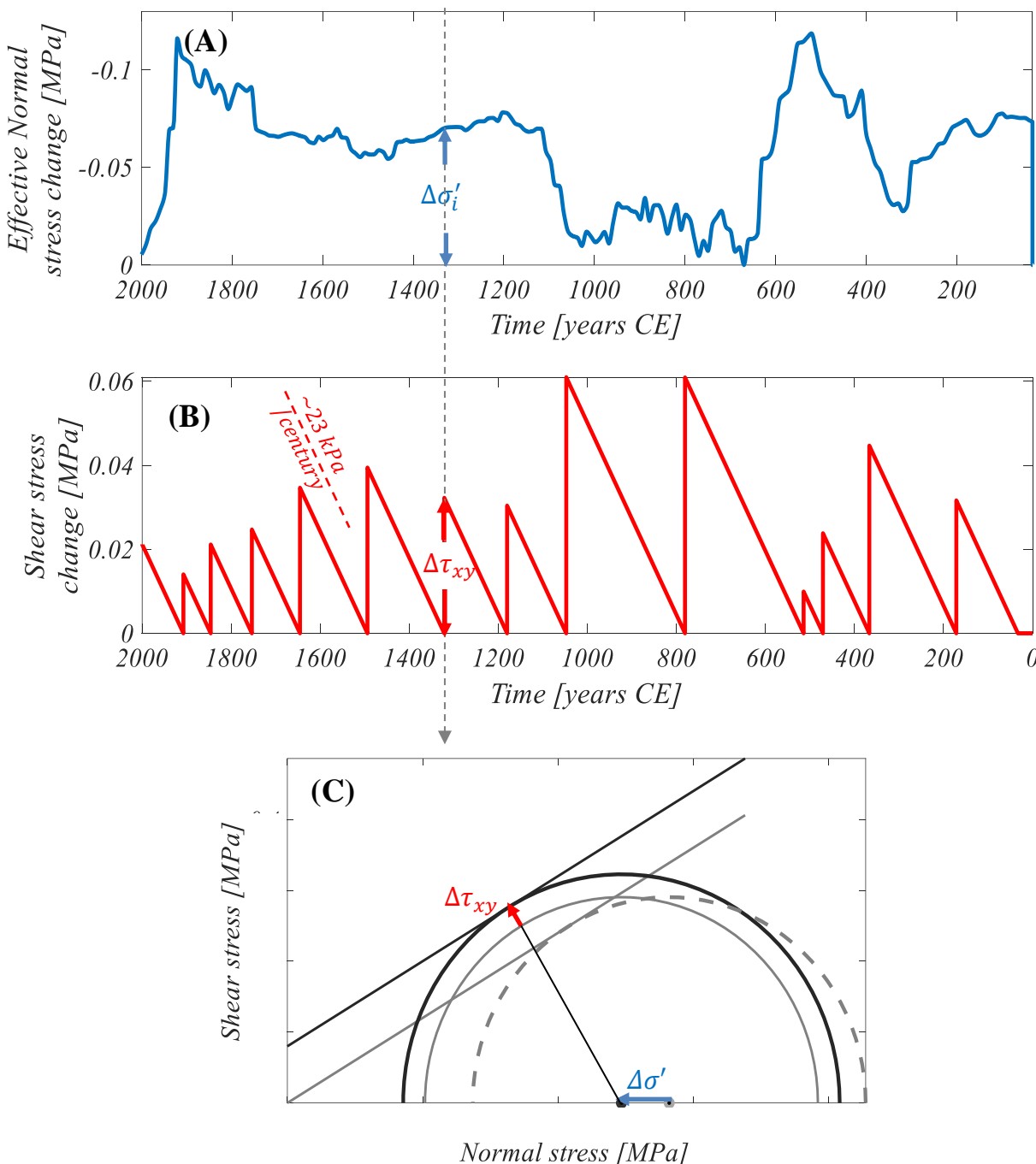

**Figure 3:** *(A) The effective normal stress change, $\Delta\sigma'$, induced by WL change, Eq.1. (B)Tectonic shear stress change, $\Delta\tau_{xy}$, accumulated consistently with slip-rate on the strike-slip fault during the time passed since the last earthquake. The shear stress accumulation rate, used in this study is about 23 kPa/century (formulation below Eq. 2, following Belferman et al., 2018) (C) Evolution of the stress change on the fault due to combined tectonic and water loading. The state of the effective stress at the fault immediately after an earthquake is restricted by the Byerlee's law envelope with zero cohesion, C=0, and a friction angle, $\varphi = 0.54^o$. The center of the Mohr circle is located at ($\sigma_0, \tau_0=0$ see Belferman et al., 2018 for more detail). The failure envelope is defined by $C\geq0$ and $\varphi = 0.54^o$. The left shift in the center of the circle by $\Delta\sigma'$ represents pore pressure (due to WL) change at this moment (Fig.3A); the increase in radius represents tectonic shear stress, $\Delta\tau_{xy}$, accumulated during the inter-seismic period (Fig.3B). Failure occurs when the circle tangents the failure envelope (presented here for the representatve 1320 CE earthquake).*

## DISCUSSION

Uncertainties in the WL reconstructions associated with dating and resolution lead to considerable variance in possible interpolations (Figure 1B). A Pearson correlation coefficient test shows that most of the randomly interpolated WL curves give linear correlation with earthquake RIs (indicated by a mean Pearson coefficient of R=0.63), excluding the three outliers (Figure 1D) to be discussed below. Figure 2 shows a similar pattern of the WL change for the ten most correlated curves. In all cases, a significant rise in the WL of about 400 CE and 1100 CE is visible and a decrease in the WL around 200 and 600 CE. Also, the maximum level around 500 and 1900 CE appears in all ten cases.

For simulating synthetic earthquakes triggered by the WL change, we use the WL curve that generates the highest correlation with the revised historical catalog (R = 0.912) (Figure 2). The dates of these simulated synthetic earthquakes are comparable with historical earthquakes (Figure 1E) excluding two events, whose date labels are offset to the y-axis for clarity of presentation (1753 CE, 1180 CE). The dates of these synthetic earthquakes might be connected to three outliers from the historical catalog (1834 CE, 1293 CE and 749 CE depicted in Figure 1D) as explained below.

The 1180 CE synthetic earthquake (Figure 1E) is comparable to an earthquake in the literature dated by Ben-Menachem (1979) and Amiran et al. (1994) to the mid-12th century (~1150 CE). Ambraseys (2009) doubted the precise dating but accepted this mid-12th century estimate. The damaged area of this earthquake spanned Jericho and Jerusalem, and the event could be considered significant because it led to the total destruction of two monasteries, one of which is 10 km south of Jerusalem's curtain wall. By admitting the ~1150 CE earthquake to the amended

catalog, we reduce the RI of the subsequent earthquake at 1293 CE (Figure 1D) from 260 to 143
yrs, thereby bringing this outlier very close to the linear correlation.

Our model also generates an earthquake in the 18th century, dated 1753 CE, for which there

were no matches in our initial historical catalog (Belferman et al., 2018). However, in Amiran's et
al. (1994) catalog an earthquake in 1712 CE is indicated: 'The quake shook the solid houses and
ruined three Turkish houses. Felt in Ramle, but not in Jaffa'. Additionally, this earthquake is
evidenced by seismites dated to 1700 – 1712 CE from an Ein Gedi site (Migowski et al., 2004).

Regarding the modeled 1907 CE event, we note the well-documented (although often

overlooked) 29 March 1903 CE earthquake (Amiran et al., 1994). This was a moderate but
prolonged earthquake: local intensity reached VII in a number of localities distributed outside the
rift valley over an area of 140x70 square km (including Jerusalem), whereas the maximum
intensity reported in the rift was VII as well (Jericho). We prefer to correlate the modeled 1907
event with the stronger 1927 Jericho earthquake that clearly released stress in the Dead Sea (e.g.
Shapira, et al., 1993; Avni et al., 2002; Agnon, 2014). This leaves the 1903 CE unmatched to our
model. Perhaps the earthquake ruptured the northern part of the central Jordan Valley, north of the
Dead Sea and south of Lake Kinneret (Sea of Galilee).

Regarding the last outlier from the historical earthquakes dated to 749 CE (or its neighbors

747 and 757, Table A1 in the Appendix) (Figure 1D) and corresponding to the simulated 780 CE
earthquake (Figure 1E): the simulation generated the preceding earthquake 514 CE associated with
the 659/660 CE event from the literature (Table A1 in the Appendix) with a deviation of 146 years.
The rupture zone of the 659/660 CE event is uncertain, and this earthquake is not necessarily
related to stress-release at the Dead Sea basin. Alternatively, following Russell (1985), as a result
of the 551 CE earthquake, a fortress east of the southern Dead Sea and Petra were destroyed.
Newer data contradicts the assertion regarding Petra; a failure in the Dead Sea region is still
plausible. Replacing the 660 CE earthquake with 551 CE in the catalog changes the RI preceding
the 749 CE historical earthquake from 89 to 198, which brings this outlier into a satisfactory linear
correlation (Figure 1D).
Additionally, it should be emphasized that in the simulation presented in this article, the
starting point is quite arbitrarily, the earthquake of 33 CE. This event together with the subsequent
earthquakes 90 CE and 112 CE (not predicted by our model) span a single century where the
catalog is nebulous. Each of these events could thus represent the starting point of the simulations
or could be omitted at this early and poorly documented interval.
Summarizing the above amendments, we add to our catalog of historical events the 551 CE,
~1150 CE and 1712 CE earthquakes, and remove 559/660 CE and 90 CE, 112 CE earthquakes
(Figure 1E). Altogether, we get 14 triggered historical earthquakes. The correlation between the
WL and RI is noticeable for the various variants of the WL curve reconstruction (Figure 4).

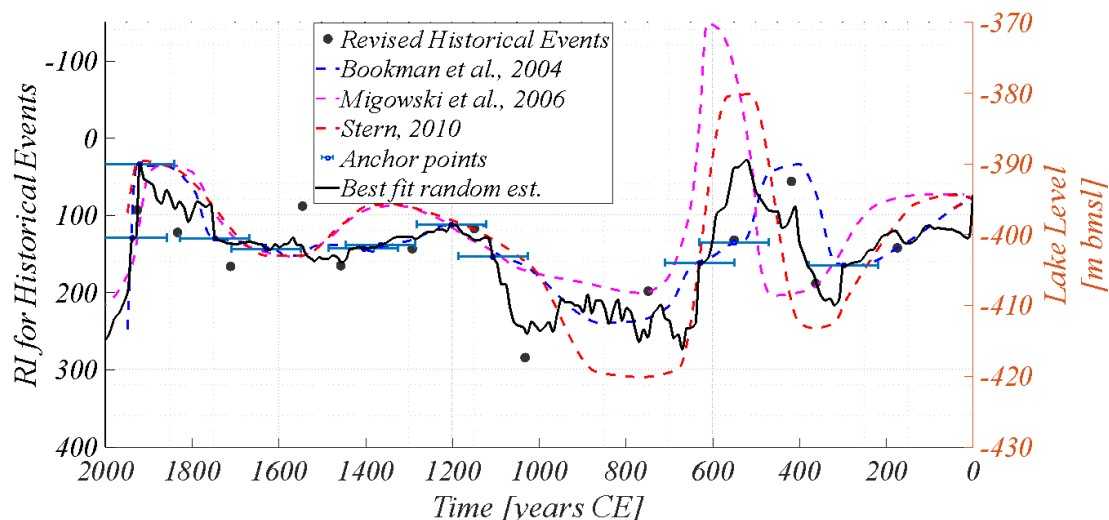


**Figure 4**: *The Dead Sea WL reconstruction for the last two millennia. The dashed curves are suggested by the literature. Blue*
*anchor points with an error interval of ±45 yr. follow Bookman et al. (2004). The solid black line is the WL curve suggested by this*
*study. The black points represent the RI for revised historical events, suggested in this study as being relevant to the Dead Sea*
*area.*

The correlation of RI with the best fit random estimated curve can be specified by a linear
prediction function:

7.                    $RI = -2483 - 6.5WL$

This linear relationship between WL and RI underscores the previously proposed
correlations between these phenomena (in Figure 9 in Belferman et al., 2018).
Since the last earthquake (1927 CE), the WL in the Dead Sea has continuously decreased at
an average annual rate of ~1 m/yr. Today the WL is about -440 (m bmsl), thus our prediction
function (Eq. 7) suggests a RI of 377 yr, for such a WL. Namely, should the WL in the Dead Sea
remain constant (-440 m bmsl), as intended in some mitigation plans, we would expect the next
earthquake at about ~2300 CE.
This paper stresses that reconstruction of WL curves is not unique and may take various
forms under the constraints available (e.g., Figure 1A). However, the correlation with an
independent record of RIs of seismic events, assuming that earthquakes are affected by WL hikes,
allows deciphering plausible scenarios for WL evolution. Moreover, for cases with the best but
not perfect correlation, the deviation might be consistent with a release of elastic energy by smaller
earthquakes, which are not accounted for by the deterministic part of our model. We note that
smaller earthquakes might rupture dip-slip fault planes, again not accounted for by our simple
model.
Additionally, as large earthquakes are accompanied by aftershocks, some of the elastic
energy is released by them. It was shown earlier that in areas of reservoir-induced seismicity,
earthquakes are not only accompanied by aftershocks but also preceded by foreshocks (Gupta,
2002). The decay curve of this kind of seismicity satisfies the criteria for the second class of
earthquake sequences by Mogi (1963). The lack of instrumental records of historical earthquakes
in our study area does not allow comparison with this class. The 1995 Gulf of Aqaba earthquake
(7.2 Mw), the last large instrumentally recorded earthquake, was accompanied by a long period
(significant enough for stress release consideration) of the aftershocks. The earthquake occurred
along the southern part of the plate boundary, which is far enough from the Dead Sea, and most
likely is not influenced by the WL change. Following this earthquake felt aftershocks continued
for about two years. At least 50 percent of the total moment associated with these aftershocks was
released during the first day after the main shock and over 95 percent in the first 3 months (Baer,
2008). In total, the post-seismic moment released during the period of 6 months to 2 yr after the
Nuweiba earthquake is about 15 percent of the co-seismic moment release (Baer, 2008). This
earthquake showed that the response of the crust to earthquakes by aftershocks is negligible, as
noted for many large earthquakes (e.g., Scholz, 1972).
For the case of artificial reservoirs, it was shown that for reservoir-induced seismicity
sequences, aftershocks continue for a longer time than for tectonic earthquake sequences (Gupta,
2002). However, given the time scale of RI, the period of aftershocks is insufficient to consider
earthquakes from the sequence in our model as separate events. Regarding the time scale presented
in our study, when the minimal inter-seismic period is about 50 years, the stress released during a
post-seismic period can be considered a part of the main shock.
The mechanical model used in this article is rather simplistic, where earthquakes release the
strike-slip component of the tectonic loading (Figure 3B). The basins around the Dead Sea fault
system also testify for an extensional component that could be manifested in co-seismic motion
along normal faults. To justify our focus on a single type of fault (strike-slip), we list the following
arguments:
• The far-field maximal and minimal principal stresses in the Dead Sea region are horizontal

(Hofstetter et al., 2007; Palano et al. 2013). This is compatible with the dominance of

strike-slip faulting (Anderson, 1951). The tectonic motion at the DSF is characterized

predominantly by a left-lateral strike-slip regime with a velocity of ~5 mm/yr along various

segments (Garfunkel, 2014; Masson et al.,2015; Sadeh et al., 2012). Large earthquakes that

initiate clusters are likely to rupture along the straight ~100 km strike-slip segments

(Lyakhovsky et al., 2001). The strike of these segments parallels the relative plate velocity

vector and thus can be approximated by simple shear. Additionally, in the Dead Sea basin,

GPS surveys indicate the dominance of strike-slip loading. Hamiel et al. (2018) show that,

on a plate scale, horizontal shear loading dominates the velocity north of the lake. Hamiel

and Piatibratova (2019) detected a sub-mm/yr component of extension across the southern

normal fault bounding the Dead Sea pull-apart (Amatzyahu Fault); yet the strike-slip

component across this very fault is much larger.

• Normal, as well as strike-slip faults, similarly react to WL change that contributes to the

vertical stress component and pore pressure change. The seismicity induced by surface WL

fluctuations and affected by the faulting regime is critically determined by the relative

orientations of the three principal stresses in the Earth's crust (Anderson, 1951). In regions

where the vertical compressive stress is not minimal (normal and strike-slip faulting),

seismic activity is more sensitive to the effective stress change due to WL change, than in
regions where it is minimal (thrust faulting) (Simpson, 1976; Snow, 1982; Roeloffs,
1988). This is applicable to reservoirs approximated as "infinite" in the horizontal plane
(e.g., Wang, 2000), with respect to the fault zone horizontal cross-section. Since we are
using a one-dimensional model, such approximation is valid for our study area where the
Dead Sea is large enough in a horizontal plane (100 km x 10 km) compared to the thickness
of the underlying strike-slip fault (cross-section) located in the central part of the valley.
Our results demonstrate that a fairly simple forward model (based on 1D analytical
solution, Belferman et al., 2018) achieves a convincing correlation between WLs and RIs of
moderate-to-strong earthquakes on the Dead Sea fault. Whereas the fault system along the Dead
Sea fault is more complicated, three-dimensional modeling of the tectonic motion, coupled with
the pore pressure evolution, may give more reliable predictions regarding earthquake ruptures and
their chronology. However, based on the relationship between the WL and RI changes presented
in this article, with the current anthropogenic decrease in the Dead Sea level (with an average
annual rate of ~ 1 m / yr), a moderate to severe earthquake will not be triggered by the mechanism
discussed here. This article not only suggests the existence of a connection between WL and RI,
but also provides additional guidance based on this connection.

## DATA AVAILABILITY

All raw data can be provided by the corresponding authors upon request.

## AUTHOR CONTRIBUTIONS

MB and AA Conceptualization; AA data collection and analysis; MB Modelling, data
visualization and results analysis; RK Validation; MB original draft preparation; MB, RK and AA
review and revisions; AA, ZB and RK Funding acquisition and Resources.

## COMPETING INTERESTS

The authors declare that they have no conflict of interest.

## ACKNOWLEDGMENTS

This project was supported by grants from the Ministry of Natural Infrastructures, Energy and Water Resources of Israel # *213-17-002,* GIF- German - Israeli Foundation for Scientific Research and Development # I-1280-301.8, and by PhD fellowships from the University of Haifa, Israel. The data for this paper were obtained with analytical and numerical modeling.

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

## 540 Appendix: The earthquake history of the Dead Sea environs

Numerous publications list earthquakes that hit the Dead Sea and its surroundings during the last
two millennia (e.g. Agnon, 2014; Ambraseys et al., 1994; Ambraseys, 2009; Amiran et al., 1994;
Guidoboni et al., 1994, Guidoboni and Comastri, 2005). In Belferman et al. (2018) we adopted
from the scores of listed events only the most destructive ones, typically causing local intensities
of VII or higher in Jerusalem. For a minimal epicentral distance of 30 km, this would translate to
a magnitude of ~5.7 or higher (according to the attenuation relation of Hough and Avni, 2011).
Table A1 lists the Dead Sea earthquakes considered for stress release across the Dead Sea basin
during the last two millennia. We used two criteria: noticeable damage in fortified Jerusalem, and
seismites in the northern Dead Sea. Our simple model simulates an earthquake time series, given
a water level curve. Eleven events from this time series correlate with events of magnitude ~6 or
more in the historical record. Yet, the model generates four events that are not included in our
original catalog. On the other hand, a single event (~660 CE) listed in Belferman et al. (2018) has
no counterpart in the simulations despite a wide range of level curves tested. All these curves are
generated by a random number generator, subject to constraints from field data. We first discuss
the four events required by the simulations one by one. Then we review the ~660 CE event along
with other historical events that were left out already in Belferman et al. (2018).
The earthquakes in Table 1 are classified according to the level of acceptance for being destructive
in Jerusalem. The nine events of **Class C** are all consensual, also used by Belferman et al.( 2018).
These events appear in all catalogs and lists and need no further discussion. The six events of **Class**
**A** are debated events, accepted in the present study. All earthquakes in this class are selected by
simultaneously satisfying two criteria: (1) The acceptance regularizes the relation between
recurrence intervals and lake level; (2) They are corroborated by evidence from seismites in the
northern basin of the Dead Sea (Ein Feshkha and Ein Gedi sites, Fig.A1corroborate).
We chose the year **33** CE to start our simulations. While this earthquake did not cause a widespread
damage, it was recorded in all three seismite sites (Kagan et al., 2011), with a maximum of decade
uncertainty based on dating by counting lamina under the microscope (Migowski et al., 2004;
Williams et al., 2012).
The second entry in Table A1, **~100** CE, refers to two decades of unrest. Migowski et al. (2004)
identified a pair of seismites around 90 CE and 112 CE in the 'Ein Gedi Core. The corresponding
sequences in Ein Feshkha and Ze'elim Creek are laminates, attesting to quiescence. A historical
hiatus between the Roman demolition of Jerusalem and the erection of Ilya Capitolina in its stead
(70-130 CE) preclude historical evidence. Although damage to the Masada fortress has been
assigned to an earthquake **1712 CE.**
Table A2 lists ten earthquakes that have been reported to damage around Jerusalem but are not
required by our simulations. The seven events of **Class R** are the debated events, rejected here
after discussion. The three **Class S** events were skipped altogether in that compilation of
Ambraseys (2009).
Of the seven Class R events, the 7 June **659** CE earthquake was accepted by us in Belferman et al.
(2018).  The earthquake has been associated with destruction of the Euthymius monastry 10 km
east of Jerusalem, but no damage in the town of Jerusalem has been unequivocally reported
(Ambraseys, 2009). In Belferman et al. (2018) we included this event in the catalog of Dead Sea
earthquakes, as Langgut et al. (2015) have located it on the center of the Jordan Valley segment of
the transform (Figure A1). However, this interpretation neglected the possibility that the rupture
could have been outside the hydrological effect of the Dead Sea basin. One of the lessons of our
numerous simulations is that our model would not support triggering of this earthquake shortly
(less than a century) before the mid-8th century crisis, when lake levels were dropping to the lowest
point in the studied period (420 m bsl, Figure 1a). When rejecting the 659 CE event, the 419 CE
earthquake is the one preceding the mid-8th century crisis; the three century recurrence interval
fits well the low lake level.
**1016** CE: The collapse of the Dome of the Rock was not explicitly attributed to an earthquake by
the original sources, who found it enigmatic as well (Ambraseys, 2009).
**1644** CE: Ambraseys (2009) quoted a late Arab author, al-Umari, who reported collapse of houses
and deaths of five persons in "the town of Filistin". While Ambraseys has interpreted it probably
to Jerusalem, it might refer to al-Ramla, the historical capital of the classical Filistin District, as in
"al-Ramla, Madinat Filastin" (Elad, 1992, p335). Or, it is a mistranslation of "Bilad Filistin" which
at that time started refer to the entire Holy Land district, without specifying a town (Gerber, 1998).
Jerusalem, at that time, was called Bayt el Maqdis or, as nowadays, al-Quds. The only report of an
earthquake in Jerusalem around 1644 mentions horror but no structural damage - the 1643 CE
event that Ambraseys (2009) tends to equate with the 1644 CE event. A seismite in Ein Gedi core
can be correlated with this event (Migowski et al., 2004, Table 2, entry 6). Migowski et al. (2004)
have identified the seismite with the 1656 earthquake that was felt in Palestine; Ambraseys' (2009)
interpretation was not yet available for them.
**1656** CE: This event was strong in Tripoli and only felt in Palestine. Migowski et al. (2004)
correlated it to a seismite based on deposition rates (no lamina counting for that interval). Given
the 1644 CE entry of Ambraseys (2009), this interpretation should be revised, and the 1656 CE
earthquake is not to be associated with any local rupture in the Dead Sea.
**Table A1:** *A catalog of earthquakes that could potentially damage Jerusalem. The classes denote the level of acceptance of*
*damage to Jerusalem among the researchers: C - consensual; B - accepted by Belferman et al., 2018; A - amended here; R -*
*rejected here.*

| Year CE or Century (marked C) | Class | Seismite correl. by site | | | Reference | Comments |
|---|---|---|---|---|---|---|
| | | ZE† | EG¥ | EF° | | |
| 33 | B | + | + | + | MI,K&,W&, | Identified in all three seismites sites, varve-counted to 31 BCE |
| 100~ | B | - | 2 | - | MI,AM | Seismites ~90 and ~112; questionable archaeologic evidence |
| ~175 | B | - | + | - | MI | A seismite; no historical or archeological support |
| 363 | C | - | - | + | K&,A& | A seiche in the Dead Sea, a seismite at EF° (north Dead Sea) |
| 419 | C | + | + | + | KT/MI/K& | |
| 551 | A | + | + | + | PA,AM | |
| 747/9,757 | C | + | + | + | KT/MI/K& | |
| 1033 | C | ? | + | + | KT/MI/K& | |
| ~1150 | A | + | - | / | AM,K& | $I_0$ IX - Mar Elias (& Qasr al-Yahud) monastries demolished |
| 1293 | C | + | + | + | K& | |
| 1458 | C | + | + | h | MI | |
| 1546 | C | / | + | i | MI | |
| 1712 | A | / | + | a | MI | A& / $I_0$ VII - "ruined three Turkish houses in Jerusalem" |
| 1834 | C | + | + | t | KT,MI | |
| 1903 | R | m | m | u | A&,AM | $I_0$ VII Mt. of Olives; several shocks, $I_0$ up to VII over a large area |
| 1927 | C | + | + | s | KT,MI | AV / $I_0$ VII-VIII in and around Jerusalem ($I_0$ 7.8 by GMPE) |


**Table A2***: Events listed in some catalogs and subsequently skipped (Class S) or declined (Class D) by Ambraseys (2009), or*
*rejected (Class R) in the present study.*

| Year CE | Class | Seismite correl. by site | | | Reference | Comments |
|---|---|---|---|---|---|---|
| | | ZE† | EG¥ | EF° | | |
| ~659 | R | - | + | + | L&,AM | Jordan Valley, possibly over 65 km NE of Jerusalem |
| 808 | S | / | - | ? | A& | |
| 1016 | D | ? | ? | ? | AM,A& | Damage to the Dome of Rock, no specific reference to shaking |
| 1042 | S | - | + | - | BM | Syria, off the Dead Sea transform |
| 1060 | S | / | - | + | A&,SB | The roof of Al-Aqsa collapsed |
| 1063 | R | | | | A&,AM,SB | Syrian littoral |
| 1068 | D | + | + | + | AM | Neither of the two events can be associated with the Dead Sea |
| 1105 | D | ? | ? | ? | A&,AM | "Strong" but "no damage recorded in the sources" |
| 1114 | D | + | + | ? | A&,AM | 1114 - no damage around the city, a swarm, Kingdom's north |
| **~1117** | R | + | | ? | A&,AM | |
| 1557 | R | | | | Am | Collapse in Jerusalem: a gun foundry, a forgery, an oven |
| 1644 | R | h | +* | h | Am | Some damage and death toll in Palestine, likely Seismite 6 of MI |
| 1656 | R | h | - | h | A&,AM,SB | Tripoli VII, Palestine IV, MI misidentified with Seismite 6 |
| 1817 | R | | | | AM | Two churches damaged in Jerusalem, Holy Sepulchre affected |
| 1870 | S | ? | - | h | AM | Mediterranean source |

 Abbreviations and notes:
†ZE - Zeʻ elim Creek; ¥EG - Ein Gedi core; °EF - Ein-Feshkha Nature Reserve
AM: Ambraseys, 2009; A&: Amiran et al., 1994; K&: Kagan et al., 2011; L&: Langgut et al.
2015; KT: Ken-Tor et al., 2004; MI: Migowski et al., 2004; PA: Parker, 1982; W&: Williams et
al., 2012.




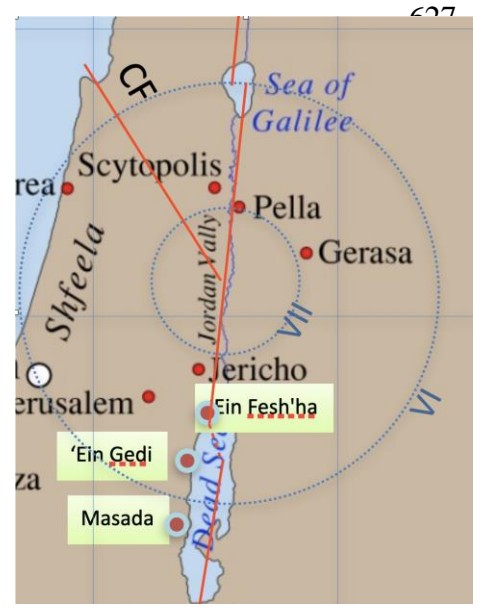

**Figure A1**: *A map showing the epicenter reconstructed by Langgut et al. (2015) for the* 659/660 CE mainshock.