# Peer review of "Identifying plausible historical scenarios for coupled lake level"

_Natural Hazards and Earth System Sciences, 2021_

## Author Response (AR1)

Dear Editor,

With utmost gratitude for reviewers' input and for editorial commitment, we list below response along with all the relevant changes made to the manuscript following the comments.

**Referee #1**

Comment 1:

- *According to the reviewer, the correlation between water level and seismic activity at the Dead Sea Basin is insignificant (our Fig.1).*

**Response:**

Our earlier work demonstrated an interesting correlation between water level and the recurrence interval of Dead Sea earthquakes (Belferman et al., 2018, Fig. 9). The present study (Figure 3 and corresponding text – pp. 16, lines 259-260 and Eq.7, line 268) indicates that the correlation between seismicity and water level change might even be higher than that demonstrated in Belferman et al., 2018 (Fig. 9). We explain that significant uncertainties in the water level data-set might hide the correlation (please see pp.3 54-76). The significance of the new correlations is borne out by the Pierson Correlation Coefficient (in excess of 0.9).

**Relevant changes:**

An explanation of these results has been added as outlined in lines 269-270.

To confirm the significance of linear correlation between WL and RI, we present Figure 2, ten best fit, randomly generated WL curves by the Pearson product-moment correlation test.

**Please see our response for the next comments. The relevant changes for all next comments of Referee #1 present under the summary response.**

- *Seismicity occurs equally at high and low levels (the reviewer refers us to Fig. 1 c and d).*

**Response:**

Our model does not preclude seismic activity during water level drops, it just takes longer for the loading to reach the increased threshold. We even predict the next future earthquake, when levels are projected to reach an unprecedented low. The reviewer writes about the correlation between the water level and seismic activity; please note - Fig. 1c, d shows that the change in level inversely correlates with the recurrence interval of earthquakes, rather than with "seismic activity". In the submitted paper the linear correlation is clearly visible in the figures, especially in Fig. 1c. The Pearson correlation coefficient reported in detail should make this point clear.

- *According to the reviewer, all change in seismic activity is due to changes in the tectonic loading and the state of the solid crust. By his assessment, the water level change has a minor effect.*

**Response:**

Tectonic motion is the primary cause and consequence of earthquakes in our model as well, but it is worth considering other factors **triggering** earthquakes. We refer the reviewer to our Introduction section: water level induced seismicity in particular, and anthropogenically induced seismicity in general, are broadly recognized phenomena over the world.

The seismic response has been observed to be connected to reservoir impoundments and their seasonal water level variations (Gupta, 1992; Talwani, 1997; Tomic et al., 2009; Hua et al., 2013).

Davies et al. (2013) compiled 198 examples of anthropogenically induced seismicity since 1929 all over the world, where higher magnitude earthquakes within this list (up to 7.9, in Zipingpu reservoir area, China 2006) are most often associated with reservoir impoundment. Foulger et al. (2018) presented a comprehensive and up-to-date database of seismicity induced or potentially induced by anthropogenic sources.

van der Elst et al. 2016 show that the appearance of induced seismicity requires a reconsideration of seismic hazards, even in regions of formerly negligible seismic concern. Simpson et al. 2018 & Gupta 2018 indicate many statistical studies that reaffirm this correlation.

In our previous work (Belferman et al., 2018), we indicated that this phenomenon may even be more applicable to the historic lakes, like those occupying the tectonic depression of the Dead Sea, due to the larger scale of water level fluctuations. The post-diffusion stage (when pore pressure at all depth approaches the value at the lake bed), is relevant for early historical data. At this post-diffusion stage, an increase in pore pressure due to the water level rise always dominates over the loading effect, leading to varying but always negative normal effective horizontal stress change at any combination of poroelastic coefficients (please see Belferman et al., 2018 Fig. 5b). This should trigger earthquakes with magnitudes higher than those in the corresponding immediate undrained (rapid) response, coming shortly after the corresponding water level rise (confirmed, for instance, for Lake Mead, Gupta, 2002; Simpson et al., 1988; Monticello reservoir, Rajendran and Talwani, 1992, and others).

 Comment 2:

- *The reviewer doubts the assumption that the effective normal vertical stress vanishes, and argues that given a reasonable permeability, it will take millions of years. In particular, he argues that between seismic events, faults heal and the permeability of the faults is very low. He points out that the Ein-Feshkha spring, very close to an exposed fault, demonstrates a lack of hydraulic connection with the deep aquifers.*

**Response:**

Pore pressure diffusion in fault zones is usually dominated by fault damage zones, which act as conduits (Simpson et al., 1988). The reviewer mentions damage processes as a possible determinant of seismicity fluctuations. We totally agree and refer to Hamiel et al. (2005) where we simulated how, under damage accumulation, slender porous zones evolve and become faults. Recently, a poroelastic analysis by Jim Rice's group of a real

case of induced seismicity considered a realistic damage/permeability structure with a fault core and anisotropic damage/permeability zones. They have shown that the permeable zones indeed propagate from shallow depths to the seismogenic zone with hydraulic fracture (Yehya et al., 2018). In fact, a useful illustration of a related phenomenon was given by Yechieli and Bein (2002), preceding the largest instrumental earthquake on the Dead Sea Transform - the 1995 Gulf of Aqaba M7.2 event some 200 km away.

Some of the material characteristics in fault zones may vary significantly, depending on the types of host rock and on state variables (e.g. temperature, pore pressure, depth, etc.). Specifically, values of hydraulic diffusivity of intact rocks differ considerably among rock types (Rice and Cleary, 1976), and strongly decay with depth (e.g. by ~6 orders of magnitude at a depth of 20 km compared to the shallow crustal values (Ingebritsen and Manning, 2010). However, such laboratory values are not determined under conditions approaching hydrofracture.

Hydraulic diffusivity around faults, as obtained from various seismological observations, is independent of earthquake hypocentral depth and estimated to be on the order of 5.0 m2/s, termed as "seismogenic diffusivity" (e.g. Talwani and Acree, 1985; Hua et al., 2013; Talwani, 1997; 2007). This suggests that the flow through faults is relatively independent of the rock type (see Ingebritsen and Manning, 2010, and references therein). Therefore, the diffusivity of faulted rock in our study is best represented by the typical value of high-porosity Boise sandstone, 4.0 m2/s (Wang, 2000), as used for our calculations.

 Using this diffusivity, we calculate the diffusion time scale, that indicates the time of arrival of the excess pore pressure from the lake's bed to hypocentral depths over the fault, for an average hypocentral depth underneath the Dead Sea fault system, z $\cong$ 20 km (Aldersons et al., 2003; Godey et al., 2006; Shamir, 2006). This time scale is ~3yr, according to the characteristic time scale definition (e.g. Kirby, 2010; Wang, 2000). For more details see Belferman et all., 2018, Eq.3 and explanation following the equation.

This time scale is negligible compared to the typical recurrence intervals of decades and longer, typical of moderate-to-large earthquakes (M > 5.5) in the Dead Sea area during the studied period (see Tables A1, A2, and text in Appendix chapter in the current ms). It is also insignificant compared to the minimal temporal uncertainty in the lake level curve, nominally taken as 10 years in our simulations.

Regarding the artesian aquifers around Ein Feshkha - these are very local, minimal (two out of numerous boreholes), shallow, and ephemeral (Swaed et al., Geological Survey Report: GSI-10-2014).  Despite the proximity to the impressive fault scarps preserved in the bedrock, the relationships between the artesian aquifers and the faults are poorly established, as is the historical activity of the nearby fault (Sharon, et al., 2020). More relevant are the thermal waters, ubiquitous in the basin. These require connectivity within the top kilometers of the section.

- *The reviewer notes that even shallow boreholes at the Dead Sea are over-pressurized. Following this, he rules out vanishing of the normal vertical effective stress.*

**Response:**

Our response concerns two points, mechanical and hydrological. The manuscript explicitly treats the effect of water level change relative to its minimal level (415 m below mean sea level - bmsl) over the period studied (pp. 9, rows 175-179). The normal stress change generated by such water level change is superimposed on the ambient regional tectonic stress field (please see pp.7 rows 131-139). First of all, we do not claim that normal vertical stress is zero. Moreover, being under the influence of gravity, the vertical stress depends on the hypocentral depth. Our model is based on the water level change effect being only on horizontal components of the normal effective stress (as was shown previously for the reservoir-induced seismicity by Simpson 1976, and then for large-scale surface water level fluctuations, in Belferman et al., Eq.10b and corresponding text in p.393). Simpson (1976) established that in all faulting environments, the vertical stress increase from water loading is eventually canceled out by increased pore pressure and the final, post-diffusion stage (as explained above), is one in which the vertical stress returns to its initial value. Secondly, assuming that the reviewer is referring to Ein Feshkha boreholes, as we noted above, their relationships with faults are poorly established, as is the historical activity of the nearby fault (Sharon, et al., 2020).

On the hydrological aspect, we expand our response above for artesian waters around Ein-Feshkha. The shallow artesian aquifers mentioned by the reviewer are of the type described by Mazor et al. (1995) for the Dead Sea rift. These are isolated and local aquifers, trapped in the fine sediments. In the Dead Sea area, such aquifers occasionally seep, as reported by sinkhole studies (e.g. Al-Halbouni et al., 2017). The fluids spewed out are described as "sediment-laden", and we interpret this to indicate suspension of fine-grain material from the shallow hydrologic system. These local effects do not necessarily represent the deeper aquifers that might be connected to the surface via fault zones as described above. According to earlier descriptions of artesian water in drill holes (Shiftan, 1958; Bentor, 1961), these wells also tap local aquifers that do not interact with active faults (at least during the last two millennia).

Comment 3:

- *The reviewer doubts the assumption that the pore pressure throughout the Dead Sea Basin at any depth responds elastically to changes in the lake level. According to the reviewer, since the basin is mostly made of clay, having plastic properties, changing the water level by 25 meters will not affect the depth of 2-3 km.*

**Response:**

With regard to the composition of the Dead Sea deposits, it was shown that clay is a minor component of the Dead Sea Basin (e.g. Frydman et al., 2008; Herut et al., 1997; Haliva-Cohen et al., 2012). Frydman et al., (2008) measured geotechnical properties and found vanishing cohesion, as opposed to any lithology dominated by clay minerals. They stated that even where the particle size is clay-size the mineralogy is not of clay. Herut et al., (1997) and Haliva-Cohen et al. (2012) characterized the mineralogies and (like others) found quartz and carbonates to be the leading detrital minerals in the fine sediments.

In summary, the reviewer seems to have misinterpreted some key points in our work. In our revision, we endeavour to better present our assumptions and results to avoid confusion. With

the present knowledge of the hydrological and lithological properties of the Dead Sea Basin, as documented by current literature (referenced), our assumptions and conclusions remain intact.

**Relevant changes:**

The arguments presented above were detailed in our previous article (Belferman et al., 2018), to which we repeatedly refer. However, for clarification in the Methods section, we have outlined the essence in more detail (pp.6-7 125-133, 140-142).

To clarify the principle of our model and results, in the methods chapter, we highlighted that tectonic stress accumulated consistently with slip-rate (pp.6 122-123 and pp.7 145-146).

In addition, please see pp. 18-19, 306-335, where we list the following arguments regarding the simplification of our model.

**Referee #2**

- *This paper by Belferman et al. analyzes the correlation between the historical record of lake levels at the Dead Sea and regional seismicity through a numerical simulation of earthquake catalogs. The numerical simulations are based on relating the seismic stress release to characteristic water level curves derived from known control points, or dates of confirmed water levels (with associated uncertainties). The authors find a high correlation between the water level changes and historical earthquake recurrence interval. Overall, the paper is well written and lays out the appropriate motivation for this study. A couple comments are presented below.*
- *As I understand, the assumption is that all stress release occurs along a single purely strike-slip fault plane. The authors acknowledge this assumption in the Discussion, and perhaps this paper is the groundwork for future modeling, but this is quite important.*

We thank the reviewer for addressing two important points, both related to stress release. We elaborate on these points in discussion in our revised manuscript.

Comment 1:

- *The distribution of smaller magnitude historical events will never be known, however using reasonable expected aftershock decay curves, one could include this additional accumulated stress release from the aftershocks in the modeling, or at least provide as a back-of-the-envelope calculation to determine its significance*

**Response:**

Large earthquakes are indeed accompanied by aftershocks, hence some of the stress is released by them. Nevertheless, the crustal response to the earthquake by aftershocks is minor and secondary compared to aseismic creep, as noted for many large earthquakes (e.g. Scholz 1972).

This holds also for the Dead Sea Transform: following the last large earthquake (7.2 Mw) that occurred in 1995 along the southern part of the plate boundary, the aftershocks continued for about two years (Fig. 1, below). At least 50 percent of the total moment associated with these aftershocks was released during the first day after the main shock and over 95 percent in the first 3 months (Baer 2008). In total, the moment released by post-seismic deformation in the period of 6 months to 2 yr after the Nuweiba earthquake is about 15 percent of the co-seismic moment release (Baer 2008).

In the time scale presented in our study, when the minimal inter-seismic period is about 50 years, the stress released during post-seismic period of 2 years can be considered a part of the main shock.

**Relevant changes:**

These arguments have been detailed in the Discussion chapter pp. 17-18 284-306.

Comment 2:

- *Secondly, the purely strike slip fault motion is likely an oversimplification of the stress release. As these events result from over-pressurized fault zones the slip distribution likely has non-double-couple components. While the total stress released is governed by the seismic moment, the length and orientation of the principal stress vectors relative to the expected shear stress can be significant for a range of plausible fault plane solutions. The modelling for this is not within scope, however my suggestion is to include some more comments regarding the strike-slip assumption.*

**Response:**

We agree with the reviewer that this aspect is beyond the scope of our research, as are other co-seismic aspects. We also accept that our assumptions regarding the strike slip orientation of the faulting process and sensitivity to the water level changes need some elaboration.

The seismologically determined far-field maximal and minimal principal stresses, in the Dead Sea region are horizontal (Hofstetter et al., 2007; Palano et al. 2013), which is compatible with the nature of the strike-slip faulting (Anderson, 1951). Under this stress field, stresses on vertical planes reach the failure criterion. In terms of Mohr-Coulonb diagrams (Fig.2, below), the large circle represents a vertical plane, on which failure should initiate.

The tectonic and interseismic motion at the DSF is characterized predominantly by left-lateral strike-slip regime with a velocity of ~5 mm/yr along various segments, supporting the seismological analysis (Garfunkel, 2014; Masson et al.,2015; Sadeh et al., 2012).

Notwithstanding, we acknowledge that due to a pull-apart basin structure of DST, it is possible that a significant part of the stress is released during motion along normal faults. For example, some of the major aftershocks and the slip resulted from 1995 earthquake, was along Gulf-parallel normal faulting NW of the main rupture (Baer 2008). In the Dead Sea Basin, GPS surveys indicate dominance of strike slip loading. Hamiel et al. (2018) show that, on a plate scale, horizontal shear loading dominates the velocity north of the lake. Hamiel and Piatibratova (2019) detected a sub mm/yr component of extension across the southern normal

fault bounding the Dead Sea pull apart, yet the strike-slip component across this very fault seems much larger.

The seismic activity induced by the surface water level fluctuations and affected by the faulting regime is determined in turn by the relative orientations of the three principal stresses (Anderson, 1951). In regions where the vertical compressive stress is not minimal (normal and strike-slip faulting), seismic activity is more sensitive to the effective stress change due to water level change, than in regions where it is minimal (thrust faulting) (Simpson, 1976; Snow, 1982; Roeloffs, 1988). This is applicable to a case of reservoirs approximated as "infinite" in the horizontal plane (e.g. Wang, 2000), with respect to the rupture area. Such an approximation is valid for our study area where the Dead Sea width is sufficiently large compared to the thickness of the underlying strike-slip fault located in the central part of the valley. Large earthquakes are likely to rupture along the straight ~100 km segments, north and south of the Basin. We expect moderate earthquakes to initiate in the basin.

These factors contribute to the high correlation observed between water level changes and historical earthquake recurrence interval.

[Figure]

Figure 1: A cumulative count of aftershock ($M_d \geq 2.4$) following the Mw7.2 Gulf of Aqaba earthquake (22/11/1995). The termination of the series is estimated by the departure of the asymptote from the data or the fit to an Omori-Utsu behavior. Agnon et al., 2021

[Figure]

Figure 2: Mohr-Coulomb diagrams for effective stresses during lateral (left) and normal (right) faulting. $\sigma_v$ denotes vertical stress, $\sigma_{Hmax}$ and $\sigma_{Hmin}$ denote maximum and minimum horizontal stresses, respectively.

**Relevant changes:**

These arguments regard the strike-slip assumption have been detailed in the Discussion chapter pp. 19 307-336.

**REFERENCE**

Agnon, A., Barnea, O., Darvasi, Y., 2021. Aftershock series duration and the time for returning to quiescence following a large earthquake. in: Planning Provisional Accommodation for Uprooted Communities; eds.: E. Feitelson, A. Agnon, E. Lederman et al., Final Report Submitted to The Ministery of Science & Technology, p. 4-27, in Hebrew.

Aldersons, F., Ben-Avraham, Z., Hofstetter A., Kissling E., Al-Yazjeen T., 2003. Lowercrustal strength under the Dead Sea basin from local earthquake data and rheological modeling. Earth Planet. Sci. Lett. 214, 129–142.

Al-Halbouni, D., Holohan, E.P., Saberi, L., Alrshdan, H., Sawarieh, A., Closson, D., Walter, T.R. and Dahm, T., 2017. Sinkholes, subsidence and subrosion on the eastern shore of the Dead Sea as revealed by a close-range photogrammetric survey. Geomorphology, 285, pp.305-324.

Anderson, E. M. (1951). The dynamics of faulting and dyke formation with applications to Britain. Oliver and Boyd.

Baer G., G. J. Funning, G. Shamir, T. J. Wright (2008). The 1995 November 22, Mw 7.2 Gulf of Elat earthquake cycle revisited, Geophysical Journal International, 175(3), 1040-1054. https://doi.org/10.1111/j.1365-246X.2008.03901.x

Belferman M., Katsman R. and Agnon A., 2018.Effectoflarge-scalesurfacewaterlevel fluctuationsonearthquakerecurrenceintervalunderstrike-slipfaulting. Tectonophysics,744,390-402.

Bentor, Y.K., 1961. Some geochemical aspects of the Dead Sea and the question of its age. Geochimica et Cosmochimica Acta, 25(4), pp.239-260.

Davies R., Foulger G., Bindley A., Styles P., 2013. Induced seismicity and hydraulic fracturing for the recovery of hydrocarbons. Mar. Pet. Geol. 45, 171–185.

Foulger GR, Wilson MP, Gluyas JG, Julian BR, Davies RJ. 2018. Global review of human-induced earthquakes. Earth-Sci. Rev. 178:438–514

Frydman S., Charrach J., & Goretsky I. 2008. Geotechnical properties of evaporite soils of the Dead Sea area. Engineering geology, 101(3-4), 236-244.

Garfunkel, Z., 2014. Lateral motion and deformation along the Dead Sea Transform. In: Garfunkel, Z., Ben-Avraham, Z., Kagan, E. (Eds.), Dead Sea Transform Fault System: Reviews. 5. Springer, Dordrecht, pp. 109–150. http://dx.doi.org/10.1007/978-94-017-8872-4.

Godey S., Bossu R., Guilbert J., Mazet-Roux G., 2006. The Euro-Mediterranean Bulletin: a comprehensive seismological Bulletin at regional scale. Seismol. Res. Lett. 77, 460–474.

Gupta H.K., 1992. Reservoir induced earthquakes. In: Gupta, H.K. (Ed.), Development in Geotechnical Engineering. 71. Elsevier, Amsterdam, pp. 1–364.

Gupta H.K., 2002. A review of recent studies of triggered earthquakes by artificial water reservoirs with special emphasis on earthquakes in Koyna, India. Earth Sci. Rev. 58 (3), 279–310.

Gupta H.,K.,2018,Reservoir triggered seismicity at Koyna, India, over the past 50 yrs. Bulletin of the Seismological Society of America 108.5B:2907-2918.

Haliva-Cohen A., Stein, M., Goldstein, S. L., Sandler, A., & Starinsky, A. (2012). Sources and transport routes of fine detritus material to the Late Quaternary Dead Sea basin. Quaternary Science Reviews, 50, 55-70.

Hamiel Y., Lyakhovsky, V., & Agnon, A. 2005. Rock dilation, nonlinear deformation, and pore pressure change under shear. Earth and Planetary Science Letters, 237(3-4), 577-589.

Hamiel, Y., Masson, F., Piatibratova, O., & Mizrahi, Y. (2018). GPS measurements of crustal deformation across the southern Arava Valley section of the Dead Sea Fault and implications to regional seismic hazard assessment. Tectonophysics, 724, 171-178. https://doi.org/10.1016/j.tecto.2018.01.016

Hamiel, Y., & Piatibratova, O. (2019). Style and distribution of slip at the margin of a pull-apart structure: Geodetic investigation of the Southern Dead Sea Basin. Journal of Geophysical Research: Solid Earth, 124(11), 12023-12033. https://doi.org/10.1029/2019JB018456

Herut B., Gavrieli, I., & Halicz, L. 1997. Sources and distribution of trace and minor elements in the western Dead Sea surface sediments. Applied geochemistry, 12(4), 497-505.

Hofstetter, R., Klinger, Y., Amrat, A. Q., Rivera, L., & Dorbath, L. (2007). Stress tensor and focal mechanisms along the Dead Sea fault and related structural elements based on seismological data. Tectonophysics, 429(3-4), 165-181. https://doi.org/10.1016/j.tecto.2006.03.010

Hua W., Chen, Z., Zheng, S., and Yan, C., 2013b. Reservoir induced seismicity in the Longtan reservoir, southwestern China. J. Seismol. 17 ( 2) 667-681.

Ingebritsen S. E., & Manning, C. E. 2010. Permeability of the continental crust: dynamic variations inferred from seismicity and metamorphism. Geofluids, 10(1-2), 193-205.

Kirby B.J., 2010. Micro- and Nanoscale Fluid Mechanics: Transport in Microfluidic. University Press, Cambridge.

Lyakhovsky, V., Ben-Zion, Y., Agnon, A., 2001. Earthquake cycle, fault zones, and seismicity patterns in a rheologically layered lithosphere. J. Geophys. Res. Solid Earth 106 (B3), 4103–4120.

Masson, F., Hamiel, Y., Agnon, A., Klinger, Y., Deprez, A., 2015. Variable behavior of the Dead Sea Fault along the southern Arava segment from GPS measurements. Compt. Rendus Geosci. 347, 161–169. http://dx.doi.org/10.1016/j.crte.2014.11.001.

Mazor, E., Gilad, D. and Fridman, V., 1995. Stagnant aquifer concept Part 2. Small scale artesian systems-Hazeva, Dead Sea Rift Valley, Israel. Journal of Hydrology, 173(1-4), pp.241-261.

Palano, M., Imprescia, P., & Gresta, S. (2013). Current stress and strain-rate fields across the Dead Sea Fault System: Constraints from seismological data and GPS observations. Earth and Planetary Science Letters, 369, 305-316. https://doi.org/10.1016/j.epsl.2013.03.043

Rajendran K., Talwani, P., 1992. The role of elastic, undrained, and drained responses in triggering earthquakes at Monticello Reservoir, South Carolina. Bull. Seismol. Soc. Am. 82 (4), 1867–1888.

Rao, N. P., & Shashidhar, D. (2016). Periodic variation of stress field in the Koyna–Warna reservoir triggered seismic zone inferred from focal mechanism studies. Tectonophysics, 679, 29-40. https://doi.org/10.1016/j.tecto.2016.04.036

Rice J.R., Cleary, P.M., 1976. Some basic stress diffusion solutions for fluid-saturated elastic porous media with compressible constituents. Rev. Geophys. 14 (2), 227–241.

Sadeh, M., Hamiel, Y., Ziv, A., Bock, Y., Fang, P., Wdowinski, S., 2012. Crustal deformation along the Dead Sea Transform and the Carmel Fault inferred from 12 years of GPS measurements. J. Geophys. Res. Solid Earth 117, B08410. http://dx.doi.org/10.1029/2012JB009241.

Scholz, C. H. (1972). Crustal movements in tectonic areas. Tectonophysics, 14(3-4), 201-217. https://doi.org/10.1016/0040-1951(72)90069-8

Shamir, G., 2006. The active structure of the Dead Sea Depression. Geol. Soc. Am. Spec. Pap. 401, 15–32.

Sharon M., Sagy, A., Kurzon, I., Marco, S., & Rosensaft, M. 2020. Assessment of seismic sources and capable faults through hierarchic tectonic criteria: Implications for seismic hazard in the Levant. Natural Hazards and Earth System Sciences, 20(1), 125-148.

Shiftan, Z.L. 1958. An artesian Aquifier of the Dead Sea. Bull. Research Council of Israel, 76, 27-52.

Simpson, D. W. (1976). Seismicity changes associated with reservoir loading. Engineering Geology, 10(2-4), 123-150.

Simpson D.W., Leith, W., Scholz, C., 1988. Two types of reservoir-induced seismicity. Bull. Seismol. Soc. Am. 78, 2025–2040.

Snow, D. T. (1982). Hydrogeology of induced seismicity and tectonism: Case histories of Kariba and Koyna. Geological Society of America Special Papers, 189, 317-360. https://doi.org/10.1130/SPE189-p317

Simpson D. W., Stachnik, J. C., & Negmatoullaev, S. K. (2018). Rate of change in lake level and its impact on reservoir triggered seismicity. Bulletin of the Seismological Society of America, 108(5B), 2943-2954.

Talwani P., 1997. On the nature of reservoir induced seismicity. Pure Appl Geophys. 150, 473-492.

Talwani P., Acree, S., 1985. Pore pressure diffusion and the mechanism of reservoir induced seismicity. In: Shimazaki, K., Stuart, W. (Eds.), Earthquake Prediction. Birkhäuser, Basel, pp. 947–965.

Talwani P., Chen, L., Gahalaut, K., 2007. Seismogenic permeability, ks. J. Geophys. Res. Solid Earth 112 (B7).

Tomic J., Abercrombie, R., Do Nascimento, A., 2009. Source parameters and rupture velocity of small M≤2.1 reservoir induced earthquakes. Geophys. J. Int. 179, 1013–1023.

van der Elst NJ, Page MT, Weiser DA, Goebel TH, Hosseini SM. 2016. Induced earthquake magnitudes are as large as (statistically) expected. J. Geophys. Res. 121:4575–90

Wang H., 2000. Theory of Linear Poroelasticity With Applications to Geomechanics and Hydrogeology. University Press, Princeton.

Yechieli, Y. and Bein, A., 2002. Response of groundwater systems in the Dead Sea Rift Valley to the Nuweiba earthquake: Changes in head, water chemistry, and near-surface effects. Journal of Geophysical Research: Solid Earth, 107(B12), pp.ETG-4.

Yehya A., Yang Z., & Rice J. R., 2018. Effect of fault architecture and permeability evolution on response to fluid injection. Journal of Geophysical Research: Solid Earth, 123, 9982–9997.

---

## Author Response (AR2)

Dear Editor,

We thank the reviewer for his useful comments and annotations regarding the manuscript, they have been considered in this revised version.

*"This is an interesting paper and I certainly recommend publication. It is speculative and I am not sure that I completely agree with all of their methods or conclusions - but that is not important in a paper like this. The ideas are well presented, the data are explained, the methods are clearly described and the rational for their conclusions are clearly stated. In searches for causative relationships between different phenomena - - in this case water level changes and earthquakes - - a single case may not be convincing, but multiple cases can build confidence in the significance (or lack thereof) of the relationship. Good documentation of case histories is important. Even when individual cases may be questionable, publication and distribution of well-presented documentation is important. Otherwise, how can common features be identified?"*

*Some specific comments:*

Comment 1:

*"It is not clear how much area is covered by the earthquakes considered in this study. Of course, since they are based on felt reports, the exact epicentral locations are difficult to determine. However, it would seem that the events considered could extend a considerable distance (100 km?) from the Dead Sea itself. It should be noted that this would be significantly farther from the lake than experienced in other cases of triggered earthquakes."*

Response:

In the present study we specifically focus on earthquakes that were reported to have toppled houses at or very near Jerusalem. The distance from the historical city to the main Jericho fault is about 30 km. The "felt" level of local intensity corresponds to $2 \leq I \leq 6$, whereas fallen houses correspond to $I \geq 7$. As shown for reservoir-induced seismicity, water level changes can generate earthquakes over very long distances from the reservoir (e.g. up to 40km Durá-Gómez and Talwani, 2010). It is explained by the diffusion along the faults. In our analysis we associate all the historical earthquakes presented (Table 1A,2A in appendix chapter) with rupture of the strike-slip faults, which agree with our modeling approach. Hence, the major strike-slip faults (Lower Jordan fault, Dead Sea Lake fault and Northern Arava fault) constituting the plate boundary could be affected by Dead Sea water level changes. Therefore, our study covers the area within

this distance. The corresponding text was included in the final version, please see lines 46-47 on pp.3 and lines 92-96 on pp.5.

Comment 2:

*It would be very useful to provide a figure showing the time history of the induced stresses, tectonic stresses and failure criteria for the synthetic earthquake catalog developed along with the water level data. This would make it easier to understand the process used to develop the link between water level and seismicity and also provide a better understanding of the relative magnitudes of the stresses involved. Without this information, I find it difficult to assess the significance of the lake induced stresses relative to the naturally occurring stresses and failure criteria.*

Response:

Please see the required figures and the corresponding explanation in the revised version of the manuscript (Figure 3, line 219 on pp.14, and corresponding text lines 212-218 on pp.13).

Comment 3:

*My oversimplification of the results of this study is that three episodes have been identified in the water level and seismicity rates - - one from 0-600 years CE with high water level and shorter recurrence intervals; the second from 600 – 1200 years CE with low water level and longer recurrence intervals; the third from 1200 – 1900 years CE with a return to higher water level and shorter recurrence intervals. In this regard, the authors should make note of Figure 5 in Ambrayses, 1971 (Nature v 232 pp 375-379, "Value of Historical Records of Earthquakes") which shows a similar cycle in the rate of seismicity. Although the Ambrayses paper is a comparison of seismicity rates between the Anatolian fault zone and the "Border Zone" (northern extension of the Dead Sea Zone), he does make the following tantalizing statement:*

*"A similar cyclic pattern, but with longer periods of overlapping activity, was noticed for the Border Zone and the Dead Sea System. At this stage, however, a more detailed study of the interaction and correlation of activity of contiguous units is not warranted." These long-term changes in seismicity rates, without a link to induced stresses, should be noted as a counter to the mechanism proposed in this paper.*

Response:

Ambraseys' paper from 1971 has guided our research for the last two decades: In Migowski et al., 2004 (cited in the manuscript), we explored (Fig. 8) Ambraseys' statement for the Dead Sea Fault. In Agnon et al., 2006 (Geol. Soc. Am. Special Paper 401, 195-214, "Intraclast breccias in laminated sequences

reviewed: Recorders of paleo-earthquakes"), we refined the picture (Fig. 13). In Agnon, 2014 (cited in the manuscript) we show (Fig. 8.17a) that the transition noted by the reviewer for 600 CE is not warranted for the entire Dead Sea fault. Yet the reviewer is correct: a transition appears in our data filtered for the Dead Sea Basin per se at 600 CE. As for the second transition, our filtered dataset indicates 1100 CE.

We have been exploring coupling across plate boundaries for some time, see e.g. Braun et al., 2011 (Israel J. Earth Sci.; 58: 257–273, "Dating speleoseismites near the Dead Sea Transform and the Carmel Fault: Clues to coupling of a plate boundary and its branch"). We find that the millennial-scale cycles, modulated by large prehistoric earthquakes, contiguous strands and branches seem to be coupled.

We are inclined to think that the coupled systems of contiguous plate boundaries are modulated by the Dead Sea level fluctuations. Why would the 100 km long Dead Sea basin affect the entire plate boundary? Likely because this unique basin is the only one where such fluctuations are permitted by the hydrogeology. However, please keep in mind that our "hard" dataset comprises only 16 points, so the results are tentative and sensitive. This limitation brought us to use a random-number generator for a kind of bootstrapping in order to test correlation between lake levels and recurrence intervals. Such an exercise for testing the correlation with the Anatolian Faults is beyond the scope of the paper. or, in Anbraseys' own words: "At this stage, however, a more detailed study of the interaction and correlation of activity of contiguous units is not warranted."